# Molecular Engines, Therapeutic Targets, and Challenges in Pediatric Brain Tumors: A Special Emphasis on Hydrogen Sulfide and RNA-Based Nano-Delivery

**DOI:** 10.3390/cancers14215244

**Published:** 2022-10-26

**Authors:** Sherif Ashraf Fahmy, Alyaa Dawoud, Yousra Ahmed Zeinelabdeen, Caroline Joseph Kiriacos, Kerolos Ashraf Daniel, Omar Eltahtawy, Miriam Mokhtar Abdelhalim, Maria Braoudaki, Rana A. Youness

**Affiliations:** 1Chemistry Department, School of Life and Medical Sciences, University of Hertfordshire Hosted by Global Academic Foundation, R5 New Capital City, Cairo 11835, Egypt; 2Molecular Genetics Research Team (MGRT), Pharmaceutical Biology Department, Faculty of Pharmacy and Biotechnology, German University in Cairo, Cairo 11835, Egypt; 3Biochemistry Department, Faculty of Pharmacy and Biotechnology, German University in Cairo, Cairo 11835, Egypt; 4Faculty of Medical Sciences/UMCG, University of Groningen, Antonius Deusinglaan 1, 9713 AV Groningen, The Netherlands; 5Biology and Biochemistry Department, School of Life and Medical Sciences, University of Hertfordshire Hosted by Global Academic Foundation, Cairo 11835, Egypt; 6Clinical, Pharmaceutical, and Biological Science Department, School of Life and Medical Sciences, University of Hertfordshire, Hatfield AL10 9AB, UK

**Keywords:** brain tumors, pediatric cancers, hydrogen sulfide, neuro-oncology, nanocarriers, nanotechnology, microRNAs, siRNAs, RNA-technology, molecular pathways

## Abstract

**Simple Summary:**

Pediatric brain tumors represent a formidable challenge in the field of oncology. Pediatric brain tumors are sub-classified into several molecular sub-types, where each one is characterized by an array of hyperactivated oncogenic molecular engines. However, there have been great efforts dedicated to portray the involved signaling pathways driving pediatric brain tumors. Yet, a full understanding of the intertwined oncogenic pathways involved is still obscure. In this review, the authors shed light on novel therapeutic targets tailored for several sub-types of pediatric brain tumors and point out the limitations of such therapeutic approaches. Hydrogen Sulfide (H_2_S) has recently been cast as an oncogenic driver in several solid malignancies, yet its role in brain tumors is still under investigation. The authors also highlight the possible involvement of H_2_S in pediatric brain tumors and propose promising brain-delivery strategies for the sake of achieving better therapeutic results for brain tumors patients.

**Abstract:**

Pediatric primary brain tumors represent a real challenge in the oncology arena. Besides the psychosocial burden, brain tumors are considered one of the most difficult-to-treat malignancies due to their sophisticated cellular and molecular pathophysiology. Notwithstanding the advances in research and the substantial efforts to develop a suitable therapy, a full understanding of the molecular pathways involved in primary brain tumors is still demanded. On the other hand, the physiological nature of the blood–brain barrier (BBB) limits the efficiency of many available treatments, including molecular therapeutic approaches. Hydrogen Sulfide (H_2_S), as a member of the gasotransmitters family, and its synthesizing machinery have represented promising molecular targets for plentiful cancer types. However, its role in primary brain tumors, generally, and pediatric types, particularly, is barely investigated. In this review, the authors shed the light on the novel role of hydrogen sulfide (H_2_S) as a prominent player in pediatric brain tumor pathophysiology and its potential as a therapeutic avenue for brain tumors. In addition, the review also focuses on the challenges and opportunities of several molecular targeting approaches and proposes promising brain-delivery strategies for the sake of achieving better therapeutic results for brain tumor patients.

## 1. Introduction

Central nervous system (CNS) tumors, including brain tumors, are considered to be at the top of the list of the most common fatal malignancies accompanied by high morbidity and mortality rates [1]. According to the American Cancer Society (ACS) statistics for 2022, the estimated number of new cases of CNS tumors is 25,050, while the estimated number of deaths is 18,280 in both genders [2]. Brain tumors cause several disabilities, including a lack of recognition, concentration, and speaking, which directly affect the quality of life [3,4,5]. More importantly, CNS tumors are considered among the most persistent and challenging tumors, with a 5-year survival rate of less than 35% [6].

Brain tumors are considered the most common solid tumors among children, accounting for almost 15–20% of childhood malignancies [7]. Statistically, the pediatric incidence rates increased from 0.5% to 0.7% between the years 2008 and 2017 [1]. Brain tumors are the leading cause of cancer-related deaths among males aged younger than 40 years and females aged younger than 20 years [8].

Children with brain tumors might suffer from nausea, stomach aches, headaches, vision loss, collapsing, and seizures due to the obstruction of the cerebrospinal fluid flow leading to an increment in the intracranial pressure [9]. Typically, brain tumors were classified based on their histology, which was the traditional way of determining the treatment and prognosis [9]. This leads to poor prognosis and treatment failures in several cases, as well as treatment-related shortcomings such as the induction of radiation neurological dysfunctions, especially at the age < 3 years old. In 2016, brain tumor classification was modified by the World Health Organization (WHO) to include molecular signaling pathways and genetic mutations [9]. For instance, the Sonic Hedgehog (SHH) subgroup of medulloblastoma manifested different cellular origins between adults and children and showed differences in somatic mutations as in human telomerase reverse transcriptase (*TERT*) promoters found mainly in adults [10,11]. Such sub-classification provided a better understanding of these groups of tumors and the approaches that should be adopted for effective treatment; such findings motivated researchers to discover novel, effective, and selective therapeutic regimens for brain tumor patients. These endeavored approaches could include immunotherapies, genetic therapies, and molecular targeted therapies [6].

Although our understanding of pediatric brain tumors has been tremendously improved throughout the years, the exact mechanisms by which these tumors arise still need to be elucidated. Yet, the currently available literature linking the aberrant expression of several oncogenic molecular pathways and brain tumors has been summarized in the current review. Figure 1 demonstrates the most common pediatric CNS tumors according to their prevalence and the associated symptoms involved in these tumors as the first indication of an abnormal condition. All indicated signs are highly associated with an increase in intracranial pressure.

The current review will emphasize the recent advances in the understanding of the molecular pathogenesis of pediatric brain tumors and shed the light on the novel role of hydrogen sulfide (H_2_S) in neuro-oncology and its possible involvement in the pathophysiology of pediatric brain tumors—hence, highlighting H_2_S as a prospective promising therapeutic target. In such a context, the recent trends in RNA-targeted therapeutics will be discussed.

This figure represents the most common types of pediatric tumors and their associated clinical picture that includes: seizures, visual problems, headache, fatigue, ataxia, nausea, and vomiting.

## 2. Different Types of Pediatric Brain Tumors and Their Molecular Subtypes

### 2.1. Medulloblastoma

Medulloblastoma is a common malignant tumor in children (composed of several molecular subtypes) dependent on several clinical aspects [12]. The four molecular subtypes unraveled by cancer genomics vary in their transcriptional profiles and include: SHH (Sonic Hedgehog), Wnt (Wingless), Group 3, and Group 4 [13,14]. The first two subgroups (SHH and WNT) are named after the pathways driving their tumorigenesis. On the other hand, controversy still exists regarding the signaling pathways underlying the other two groups and their generic names, “Group 3 and Group 4” [13,14].

#### 2.1.1. Molecular Subtypes of Medulloblastoma

##### SHH Subtype

The SHH subtype is characterized by the overexpression of genes involved in the SHH pathway. Hedgehog (HH) is a secreted protein that binds to the patched (Ptch1) receptor inhibiting Ptch1 from restraining the protooncogene Smoothened (SMO) [15]. This allows SMO to activate the target genes of the SHH pathway by stimulating the activation of GLI family zinc finger 1 (GLI1). GLI1 is then translocated to the nucleus, where it activates the oncogenes *MYCN*, cyclin D1 (*CCND1*), and cyclin D2 (*CCND2*) [16,17,18]. Negative regulators of this pathway include the suppressor of fused (SUFU) and Ptch1. These two regulators are often mutated in medulloblastoma patients, where SUFU mutations are more common in children, leading to the activation of oncogenic GLI1 [10,19,20,21,22].

##### Wnt Subtype

The Wnt subtype is associated with the most favorable prognosis, representing approximately 10% of all medulloblastoma cases common in older children and teenagers [23,24]. The binding of the Wnt ligand initiates the pathway to the Frizzled serpentine receptor (Fz) and low-density lipoprotein (LDL) forming a complex, which stimulates the phosphorylation of cytoplasmic protein Dishevelled (Dsh), which inhibits GSK3-β enzyme function [25]. This leads to the inhibition of the β-catenin destruction complex, increasing its cytoplasmic levels [26]. β-catenin is then translocated to the nucleus, where it activates transcription factors and T-Cell and lymphoid enhancer factors, stimulating the expression of WNT target genes such as *MYCN*, *AXIN2*, and *CCND1* [27,28,29]. Mutations in the β-catenin gene (*CTNNB1*) are often observed in medulloblastoma, which is the reason behind the deregulation of the Wnt pathway, which eventually leads to the increase in β-catenin levels and subsequent activation of genes that promote cellular proliferation, migration, and survival [30]. Other studies have revealed that mutations in *CTNNB1* and the accumulation of β-catenin are of good prognostic value for medulloblastoma patients [24]. Likewise, a recent in vitro study found the ectopic overexpression of β-catenin had a regressive effect on the cell growth of different medulloblastoma cell lines by reducing the number of colonies formed [31]. Additionally, the brains of adult transgenic mice expressing human β-catenin genes showed no tumor growth or morphological changes, suggesting that the *Wnt* gene has limited effects on inducing medulloblastoma in adult CNS [32]. The association between the Wnt pathway and medulloblastoma still needs to be investigated, and further research to determine whether targeting the pathway is of clinical value [33].

##### Group 3 and Group 4

Group 3 and Group 4 medulloblastomas are associated with the poorest prognosis among all other medulloblastoma subtypes. Signaling pathways driving the CNS tumorigenicity of Group 3 and Group 4 medulloblastomas still require further study [25]. The amplification of transcription factor OTX2, along with the *MYC* oncogene in Group 3, might have a role in orchestrating tumorigenicity, as the latter is involved in the differentiation of different CNS progenitors and upregulating cell cycle genes, as well as stimulating TGF-β signaling [34,35,36]. A study by Antoine et al. suggested that the deregulation of Erb-B2 Receptor Tyrosine Kinase 4 (ERBB4)-SRC signaling might be the reason underlying the pathogenesis of Group 4 medulloblastoma, where the authors found increased tyrosine kinase phosphorylation, predominantly ERBB4, and the increased expression of SRC and its downstream targets [37]. However, there are still no validated data regarding the signaling pathways driving the occurrence of Group 3 and Group 4 medulloblastomas [25].

### 2.2. Gliomas

#### 2.2.1. High-Grade Gliomas

The Wnt pathway has also been linked to driving tumorigenic features of gliomas and inducing the expression of an epithelial-mesenchymal transition (EMT) regulator: Slug and Snail (SNAI2/1) transcription factors [38]. Analyzing the components of the Wnt pathway owing to its tumorigenic effects is an ongoing area of research. The overexpression of β-catenin in high-grade gliomas is a possible outcome of the aberrant activation of the canonical Wnt/β-catenin pathway, which in turn leads to the activation of CCND1 and MYC in gliomas [39]. An in vitro study found that knocking down β-catenin in human glioma cells hindered the ability of cells to proliferate and induced apoptosis, slowing down tumor growth [40,41,42,43]. Recently, another study has also demonstrated that a non-canonical Wnt ligand known as Wnt5a is a key regulator of human glioma stem cells and stimulates their infiltration and migration capacities [44].

The epidermal growth factor receptor (EGFR), the platelet-derived growth factor receptor (PDGFR), and the MET factor are the most commonly deregulated receptor tyrosine kinases (RTKs) associated with glioma [45]. Additionally, these pathways affect other core regulators and downstream effectors of growth factor signaling as P13K and cyclin-dependent kinase inhibitor 2A (CDKN2A) [46]. These RTKs are complex to target, and resistance is a common issue encountered when targeting these receptors; not to mention, the exact mechanisms by which they induce CNS tumorigenicity and drive resistance still require intensive investigations [46].

B7-H3 (CD276) is a transmembrane protein expressed on the surface of B and T cells and is considered an essential immune checkpoint regulator [47]. This protein is highly expressed in gliomas, predominantly in high-grade gliomas, and is associated with increased cellular proliferation and invasion in human glioma cell lines and mouse models, possibly through the inhibition of the regulators of the JAK2/STAT3 signaling pathway, the suppressor of cytokine signaling 3 (SOCS-3), and Src homology 2 [SH2]-containing phosphatase-1 (SHP-1), eventually leading to the activation of the JAK2/STAT3 signaling pathway [47]. The STAT3 signal can then induce EMT by activating SNAI1 and reducing E-cadherin levels, promoting cells’ invasiveness and migration [47].

One type of high-grade glioma, which is seen exclusively in children and is considered the leading cause of death in children with brain tumors, is diffuse intrinsic pontine glioma (DIPG) [48]. Regardless of its aggressiveness and the absence of an effective treatment, DIPG originates in the pons of the brainstem [49]; however, its five-year survival is <1%, with 90% of children passing away within two years of diagnosis [48,50]. Around 80% of DIPG cases harbor a histone H3K27M mutation, which in turn leads to the loss of H3K27 trimethylation at the lysine residue of histone [50,51,52].

#### 2.2.2. Low-Grade Gliomas

Genomic sequencing allowed the discovery of some possible reasons associated with low-grade glioma; among those reasons is the aberration in the *BRAF* gene, causing the upregulation of the RAS–mitogen-activated protein kinase (RAS/MAPK) pathways [53,54,55]. In patients with Neurofibromatosis type 1 (NF-1), RAS activation is attributed to the deactivation of the Ras-GTPase activating protein neurofibromin, leading to tumors in the optic pathway (also known as optic gliomas) [56]. While in non-NF-1 patients, tumorigenicity is most commonly induced by the fusion of BRAF protein with KIAA1549 protein, this subsequently leads to the diminished regulation of BRAF and the activation of MAPK [57]. Further adding to the complexity of brain tumors is that low-grade gliomas vary molecularly between adults and children, where TP53 mutations are commonly found in adult low-grade gliomas but are not found frequently in children [58].

Recent studies have highlighted the intricate interplay between protein programmed death-1 (PD-1) and the programmed cell death ligand (PD-L1) in the pathogenesis of low-grade gliomas where various receptors such as the EGFR, Toll-like receptor (TLR), and interferon-alpha (IFN-α) receptor induce the expression of PD-L1 [59,60,61,62]. The interaction between PD-L1 and PD-1 inhibits immunomodulatory cells where T-cell responses and lymphocyte cytotoxic activity are downregulated [62,63,64]. However, the components and the exact mechanism by which the pathway stimulates CNS tumorgenicity are not known. Moreover, targeting the PD-1/PD-L1 system may be of potential therapeutic importance [62,63].

### 2.3. Ependymoma

This type is a neuroepithelial malignancy that accounts for 8–10% of all CNS tumors in children and is the third most common tumor in the age spectrum < 5 years old [29,42]. Ependymomas can affect both adults and children and can attack different locations, such as the supratentorial brain, spinal cord, and posterior fossa [27]. The most suspected site for this tumor in children is intracranially in the posterior fossa [43,44,45], leading to vomiting, ataxia, papilledema, multiple cranial nerve palsies, and increased intracranial pressure accompanied [46]. Upon the 2016 classification of the WHO, this tumor was sub-classified into four subtypes depending on histopathological features: subependymoma, classic ependymoma, myxopapillary ependymoma, and anaplastic ependymoma, as well as RELA fusion-positive ependymomas, which were also included as a subcategory to this classification [65].

Signaling pathways responsible for ependymoma are not well studied, probably due to the heterogeneity of the disease and the small cohort of patients in each subset [66]. However, there are some previously reported possible pathways that may contribute to ependymomas. For instance, intercellular tyrosine kinase signaling and Notch signaling are commonly observed in patients with ependymomas [66,67]. Both pathways have roles in the maintenance of neural stem cells. Additionally, ErbB2 (a downstream target of Notch) is often overexpressed in ependymomas and is linked to increased proliferation and poor prognosis [68].

The aberrant gene expression of members of the Wnt pathway was also observed in ependymomas. However, the common mutations in β-catenin observed in medulloblastoma patients are not found in ependymomas [69]. Hence, the exact mechanism leading to the deregulation of the Wnt pathway in ependymomas is not known [69]. SHH signaling might also have a role in mediating the tumorigenicity of ependymomas due to the upregulation of GLI1-2 and the overexpression of the insulin-like growth factor-2 (IGF-2) ligand, one of the targets of SHH [70]. Insulin-like growth factor binding proteins (IGFBPs) 2, 3, and 5 were also upregulated in ependymoma, suggesting potential interplay between the SHH and the insulin-like growth factor (IGF) signaling pathway [66,70,71]. It is also worth mentioning that some signaling pathways are known to be linked to certain types of brain tumors as indicated. Yet, the exact members of these pathways and potential key regulators leading to tumorigenicity still need to be identified.

## 3. Conventional Therapeutic Approaches

CNS cancer treatment relies mainly on surgical resection followed by radiotherapy; however, the use of the latter is still controversial for children under three years of age [7]. Survival rates vary depending on the surgical resection extent; for instance, total resection leads to 66–75% survival rates [72]. These survival rates could reach 80% if radiation is applied directly after total resection [73].

Table 1 demonstrates the drawbacks of the current methodologies adopted for treating CNS tumors. After the WHO classification in 2016 and a better understanding of each subgroup on the molecular and genetic levels, new trials were conducted to modify patients’ treatment plans. Although some of these new trials improved the outcome relatively, some of these trials failed. Table 2 also highlights the research gaps in the treatment approaches before and after 2016.

## 4. The Role of Hydrogen Sulfide in Brain Tumors

Recently, hydrogen sulfide (H_2_S) has emerged as a vital oncogenic mediator in several solid malignancies including prostate, breast, melanoma, colorectal cancer, ovarian cancer, thyroid cancer, and oral squamous cell carcinoma [84,85,86]. Yet, few studies reported the regulatory role of H_2_S in brain tumors; thus, more focus should be given to studying the critical role of H_2_S in neuro-oncology. In the following sections, the currently available literature regarding H_2_S in brain physiology and pathophysiology will be discussed for deeper visualization of its potential involvement in brain tumors.

### 4.1. Biosynthesis of H_2_S

Physiologically, H_2_S is synthesized intracellularly by three chief enzymes: cystathionine β-synthetase (CBS), cystathionine γ-lyase (CSE), and 3-mercaptopyruvate sulfurtransferase (3MST). Usually, the three enzymes help the human body remove the toxic homocysteine [87]. Both CBS and CSE use L-cysteine amino acid as a precursor for H_2_S synthesis, while 3MST synthesizes H_2_S using the 3-mercaptopyruvate (a product of L-cysteine metabolism by Cysteine aminotransferase (CAT)) [88]. In the past, H_2_S was reported as a central neurotoxic agent [89]; however, recent studies have unraveled its critical physiological role in curing several CNS-related diseases [90]. For instance, the anti-inflammatory and neuroprotective effects of H_2_S generated by CBS have been reported [91]. Nevertheless, the spatial distribution of CBS, CSE, and 3MST in the brain and their correlation with brain tumors are still unclear and should be investigated further to understand their contribution to brain oncogenesis [88].

### 4.2. Cystathionine β-Synthetase (CBS)

#### 4.2.1. Genetic Location

CBS is encoded on chromosome 21 (21q22.3) [92,93] and consists of 17 exons, the first two of which belong to the 5′ untranslated region (UTR), while the third exon has the translation start site, ATG, whereas 3′ UTR is located in the 7th exon (Figure 2A) [92,93,94].

#### 4.2.2. Structure

The CBS gene expression results in a polypeptide consisting of 552 amino acids with a molecular weight of 63 kDa [95]. However, shorter isozymes (≈48 kDa) were extracted from the liver and were linked to the higher catalytic activity of the enzyme [96]. Typical CBS consists of tetramers and has been reported to have three architectural domains: the N-terminal heme-binding domain (residues 1–70), the conserved catalytic domain (residues 70–386), and the C-terminal regulatory domain (residues 386–551) that contains an allosteric cleft in which S-Adenosyl-methionine (SAM) binds and activate the CBS protein [87,95,97]. In the N-terminal domain, heme binds transiently to Cys15, which is part of the intrinsically disordered region (IDR) (residues 1–40), then binds axially to residues of Cys52 and His65 and is anchored in a hydrophobic pocket [98]. Intriguingly, heme is not involved in the catalytic activity of CBS; however, it contributes to CBS folding and is also thought to influence CBS activity due to heme’s sensitivity to environment redox status [87]. On the other hand, CBS catalytic activity is known to be pyridoxal 5′-phosphate (PLP)-dependent. In addition, the PLP cofactor is found to be bound to the Lys119 residue in the catalytic domain (Figure 2B) [98].

#### 4.2.3. Area of High Expression

CBS spatial distribution in different brain areas was investigated in various species, including mice and humans. In one study, the screening of CBS levels in the brain tissues of 3-month-old CBS(+/+) mice compared to CBS(−/−) mice revealed that CBS expression was high in the Purkinje cell layer in the cerebellum and CA2 and CA3 neurons in the hippocampus tissues. In contrast, a weaker expression of CBS was found in the CA1 neurons and the dentate gyrus (Figure 2C) [99]. In addition, lower expression levels were observed in the neurons of the granular cell layer of the cerebellum [99]. These findings were confirmed by Western blotting, in situ hybridization, and northern blotting.

On the other hand, in the human brain, CBS was found to be widely expressed in all brain and spinal cord areas [91,100,101,102]. In an immunohistochemical study on the brain and spine of eight men aged 18–44 years old who died from reasons unrelated to the CNS, CBS expression was positive in the medulla tissue pons midbrain and the spinal cord [101]. However, CBS content was variable among motor and sensory nuclei [101]. Additionally, Lee et al. reported that CBS is mainly expressed in the astrocytes [91]. They have found the H_2_S production from astrocytes to be 7.6-fold higher than that of microglial cells [91,100].

#### 4.2.4. CBS Screening in Brain Tumors and Its Significance in Disease Progression

Despite the remarkable number of studies that link H_2_S disturbances and oncogenesis, few studies are dedicated to identifying the expression level of CBS in human brain tumors and its association with disease progression [85,88,103]. However, Takano et al., have shown that the silencing of CBS gene expression by transfecting U-87 MG cells with lentiviral vectors encoding shRNA that target CBS has fastened the initiation of rapid tumor growth after xenograft subcutaneous injection but did not affect the in vitro proliferation [104]. In silico analysis explained this observation by showing that the downregulation of CBS in glioma is correlated to the increase in HIF2α protein levels, increasing oncogenic tissue behavior [104]. The targeting of CBS has been a tremendous challenge in several contexts, including several solid malignancies, such as brain tumors. Recently, several pharmacological inhibitors, as well as small and long non-coding RNAs (ncRNAs), have been reported to trim CBS activity in several cancer cell lines, as shown in Table 3. Yet, an efficient targeting system for CBS is not well-characterized and thus needs further investigations.

### 4.3. Cystathionine γ-Lyase (CSE)

#### 4.3.1. Genetic Location

CSE protein is encoded in the CTH gene on chromosome 1 (i.e., 1p31.1) as shown in Figure 3A [95]. It consists of 12 exons, whose first exon contains the 5′untranslated region (UTR) and ATG start codon while the last exon encodes for the TGA stop codon and the 3′UTR [95]. However, a shorter transcription variant was reported to lack exon 5 in its coding region [112]. Upon transfection experimentation, the longer isoform showed a 1.5-fold increase in the CSE activity, while the shorter isoform did not result in any alteration [113].

#### 4.3.2. Structure

CSE is a 405-residue polypeptide of 45 kDa whose catalytic activity is also PLP-dependent [95]. The enzyme has a tetrameric structure; each monomer consists of two domains. The N-terminal PLP-binding domain is larger (residues 9–263), and the C-terminal domain is shorter (residues 264–401) [114]. Two active sites are formed at the interface between the A/B and C/D subunits, resulting in two functionally stand-alone dimers [114,115]. At the active site, PLP is fixed by hydrogen bonds with Gly90, Leu91, Ser209, and Thr211 in subunit A and Tyr60 and Arg62 in subunit B. PLP was also found to bind to Lys212, while Tyr114 has shown aromatic interaction with the PLP pyridine ring (Figure 3C) [114].

#### 4.3.3. Area of High Expression

CSE spatial distribution in the human brain is also not well investigated. Its expression in astrocytes was limited and much lower than other H_2_S-synthesizing enzymes, CBS, and 3MST [100]. However, in a study performed by Wróbel et al., in which they investigated CSE distribution in postmortem brain tissues of human subjects aged between 20–60 years old, it was found that CSE is relatively expressed to a greater extent in the cerebellum, hypothalamus, and parietal cortex with almost the same extent. Yet, CSE was lower in the thalamus and hippocampus (Figure 3B) [116].

#### 4.3.4. CSE Screening in Brain Tumors and Significance in Disease Progression

In different grades of human brain gliomas, CSE was consistently expressed at a higher level than its usual expression in normal brain tissue [116]. The expression level of CSE increases at grade II and reaches its maximum level at grade II/III, then decreases gradually at higher tumor grades [116]. Another study on C6 glioma cells has shown that JNK and p38MAPK mediate CSE increase as a compensatory mechanism for glutathione depletion [100]. Along this line of reasoning, Cano-Galiano et al. have revealed that propargylglycine (PAG) inhibition re-sensitizes IDH1-mutant astrocytomas to cysteine depletion and diminishes tumor growth in orthotopic xenografts in mice [117]. On the other hand, CSE was also found to be involved in glioblastoma resistance to anticancer drugs such as Temozolomide [118]. This resistance was reverted upon co-treatment with Erastin, which reduced CSE activity and consequently increased Temozolomide cytotoxicity, highlighting the possible involvement of CSE in chemotherapeutic resistance in brain tumors [118]. Several approaches were probed for targeting CSE in several contexts, as shown in Table 4, but in the same scenario as CBS, an efficient targeting system is still lacking.

### 4.4. Mercaptopyruvate Sulfurtransferase (3MST)

#### 4.4.1. Genetic Location

The human 3MST enzyme has been found to be encoded by the *MPST* gene on chromosome 22 (22q13.1) [125]. The *MPST* gene is 5.6 kbp in length [126]. It has two exons; exon 1 is 595 bp, and exon 2 is 299 bp; both are separated by a large intron of about 4.4 kbp length (Figure 4A) [126].

#### 4.4.2. Structure

3MST is s a Zn-dependent enzyme in a monomer or dimer state [127]. Its tertiary structure consists of N-terminal and C-terminal domains [126]. Briefly, 3MST isoform 1 is reported to be dominant in the cytosol, with 37 kDa in weight [128]. The N-terminal domain of 3MST isoform 1 is located between residues 20–165. The first 25 amino acids in this domain are called the mitochondrial targeting sequence (MTS). Briefly, 3MST isoform 2, which is detected in both human cell cytosol and mitochondria, with 33 kDa, was found to be shorter in length through lacking the 20 amino acids that present upstream of the MTS sequence; however, both isoforms were reported to show similar kinetics in the un-physiological conditions (Figure 4B) [128,129]. The catalytic site exists in the C-terminal domain, with Cys268 or Cys248 residue that acts as a persulfide carrier in isoforms 1 and 2, respectively [128]. Likewise, cys247 residue represents the catalytic site in rat 3MST [126,127,130]. Moreover, these cysteine moieties act as an internal redox-sensing center, switching the enzyme on or off [127]. Of note, the *MPST* human gene was reported to exhibit a lot of genetic polymorphism in the intronic sequences; this, in turn, leads to different catalytic activities of the 3MST enzyme, which is mostly reduction [127].

#### 4.4.3. Area of High Expression

The importance of 3MST’s role in the brain was only highlighted after observing a significant normal production of H_2_S in the brain homogenate of CBS-knockout mice, giving that CBS was believed to be the key, if not only, H_2_S-synthesizing enzyme in CNS [131,132]. Although 3MST is reported to be remarkably expressed in brain tissues, its specific spatial distribution in the human brain is poorly investigated so far [130]. However, immunohistochemical analysis of mice and rats’ brains showed that 3MST is mainly localized in the olfactory bulb (e.g., glomerular, mitral, and external plexiform cell layers), neocortex (e.g., the cerebral cortex), hippocampus (e.g., CA1 and CA3 pyramidal cells), cerebellum (e.g., Purkinje cell somata and proximal dendrites), the pons (e.g., pontine nuclei), and in the spinal cord (e.g., large neurons) as well (Figure 4C) [131,132]. On the other hand, analysis of postmortem brain tissues of human subjects aged between 20–60 years old showed that among the hippocampus, hypothalamus, thalamus, cerebellum, frontal cortex, parietal cortex, and basal ganglia, the highest 3MST expression was found to be in the thalamus, hypothalamus, and basal ganglia in a descending order respectively (Figure 4D) [116].

#### 4.4.4. MST Screening in Brain Tumors and Its Significance in Disease Progression

Due to the relative emergence of 3MST in brain physiology and brain tumor, Wróbel et al. have investigated 3MST expression in human brain glioma tissues. It has been proposed that higher-grade brain gliomas showed lower expression levels of 3MST; however, only grades II and III have shown expression levels of 3MST higher than that found in the normal tissues [116,127].

In another study by Jurkowska et al. on the expression levels of CSE and 3MST in the astrocytoma U373 and neuroblastoma SH-SY5Y cell lines, higher expression of 3MST than CSE was reported, suggesting that 3MST is the major catalyzing enzyme of the sulfane sulfur pathway in neoplastic cells [133]. Another study by Wróbel et al., in 2017, on the expression levels of CBS, CSE, and 3MST in human neuroblastoma SHSY5Y cells and glioblastoma-astrocytoma U87 MG cell line also found 3MST with the highest levels of expression among the three investigated enzymes [134]. When it comes to 3MST, it is extremely scarce to find a specific potent pharmacological inhibitor or any other targeting approach to harness 3MST activity, leaving us with only two pharmacological inhibitors with non-specific and moderate potency as listed in Table 5.

## 5. Modern RNA-Based Therapeutic Modalities

Given the promising involvement of H_2_S in brain tumor pathophysiology and the importance of investigating it as a potential therapeutic target, the targeting of H_2_S has been quite challenging for a lot of researchers. H_2_S possesses a bell-shaped pharmacological character in different oncological contexts [85,86]. Thus, the pharmacological inhibitors of H_2_S-synthesizing enzymes did not show consistent outcomes, due to selectivity and dose- and time-adjustment issues [85]. Accordingly, molecular genetics therapeutic tools such as microRNAs (miRNAs), short hairpin RNAs (shRNAs), and small interfering RNAs (siRNAs) provide promising approaches that would allow for the specific targeting of H_2_S-synthesizing enzymes [79,84,132,133,134,135,136,137,138,139,140,141,142,143,144,145,146,147,148,149,150,151,152]. The following part of the review focuses on the possible mechanisms through which such RNA-based molecular genetics targeting approaches could be used and delivered to brain tumors.

### 5.1. Targeted Nano-Delivery of Therapeutic RNAs in Brain Tumor Therapy

Therapeutic RNAs (miRNA, siRNA, shRNA, etc.) have been used recently as novel and immuno-compatible therapeutic approaches for treating brain tumors [153]. Despite the promising suppressive activities of therapeutic RNAs to brain tumors, numerous hurdles exist that hamper their clinical translation, such as limited stability at physiological temperature, premature hydrolysis by the endogenous endonucleases found in the systemic circulation before reaching their target, and non-selective targeting [154]. More importantly, physiological barriers, including the blood-brain barrier (BBB) and the blood–tumor barrier (BTB), prevent the delivery of therapeutic RNAs to brain tumor cells [3]. Thus, suitable nano-platforms are designed to shield therapeutic RNAs from premature enzymatic hydrolysis and improve intra-cellular accumulation and selective targeted delivery. In addition, the use of nanocarriers is reported to enhance the passage of the loaded RNAs across the BBB and BTB. Several modern nano-platforms are reported as promising carriers for different RNAs, including liposomes, polymeric nanoparticles (NPs), chitosan-based NPs, supramolecules (cyclodextrin), and lipid-based NPs [155]. Numerous parameters should be considered to achieve a successful brain delivery, including size, zeta potential, and encapsulation efficiency [156]. For instance, several studies reported the influence of nanoscale size (100–300 nm) on increasing the penetration of loaded cargos across the BBB and BTB [157,158]. Additionally, it was found that positively charged NPs are favored because they can better interact with negatively charged cell membranes [159]. Despite the promising therapeutic effects of loading therapeutic RNAs into nanocarriers, few recent studies considered such innovations. In the following section, we will present some selected recent studies that reported the loading of various therapeutic RNAs into nanocarriers to improve brain tumor therapy.

### 5.2. Therapeutic RNAs Loaded Nanocarriers in Brain Tumors

Nanocarriers achieve targeting to cancer cells either via passive or active targeting. Passive targeting depends on the enhanced permeability and retention (EPR) effect, while active targeting relies on the decoration of the NPs’ surface with targeting moieties that can interact with overexpressed receptors on the surface of brain tumor cells.

Liposomes, globular nanovesicles with an aqueous interior core surrounded by phospholipids bilayers, are proposed as one of the promising nanocarriers in RNA molecule delivery to brain tumors due to their biocompatibility, ease of fabrication, high transfection capacity, and preferential accumulation into tumor tissues via the EPR effect [160].

A study reported the fabrication of multifunctional, dual-loaded cationic liposomes with impermeable vascular endothelial growth factor (VEGF)-siRNAs and chemotherapeutic docetaxel (DTX) for treating brain glioblastoma [161]. In addition, the surface of the engineered liposomes was decorated with two receptor-specific and cell-penetrating peptides (Angiopep-2 and neuropilin-1 receptor), which increased the binding capability to glioblastoma cells. The designed multifunctional liposomes were found to suppress viability and angiogenesis of brain glioblastoma cells via enhancing *VEGF* gene silencing and inducing apoptotic pathways. Besides, the engineered liposomes were found to down-regulate HIF-1α expression and CD31-positive tumor vessels and hence modulate the tumor microenvironment. Furthermore, it was revealed that the receptor-mediated transcytosis mechanism was involved in transporting the loaded cargo across the BBB and BTB. This work demonstrated the influence of liposomes as a possible nano-vehicle for the co-delivery of siRNAs and anticancer drugs in brain cancer treatment due to their safety and ability to pass through the BBB and BTB effectively [161].

Another study reported the design of liposomal nanoplatforms loaded with ferritin heavy chain 1 (FTH1)-siRNAs to treat radioresistant glioblastomas. The developed liposomes were shown to increase the activity of lactate dehydrogenase enzyme and induce the caspase 3/7, leading to mitochondrial damage and the lower cell viability of cancer cells. In addition, a remarkable radiosensitivity enhancement was observed upon radiation exposure, verified by compromised DNA repair and reduced colony formation [162].

A group of researchers used combinatorial approaches to produce shRNA–loaded liposomal formulations (100 nm) decorated with an Asn-Gly-Arg (NGR) peptide-targeting ligand that targets CD13 receptors (overexpressed on the surface of glioma cells). In addition, local ultrasonic waves were involved in improving the BBB and BTB penetration. It was revealed that the designed liposomal system extended the rat survival rate and increased the shRNA delivery to rat gliomas by about nine-fold [163].

Polyamidoamine (PAMAM) dendrimers afford another promising nanocarrier for the targeted delivery of therapeutic RNAs. PAMAM dendrimers are cationic, hyperbranched polymeric nanostructures with surface functional groups. In addition, PAMAM dendrimers can release their cargo in a pH-dependent manner due to the protonation of amino groups in the acidic microenvironment of cancer cells leading to controlled drug delivery [164].

A recent study reported the design of β-cyclodextrin (β-CD)-coated PAMAM-entrapped gold NPs to deliver two types of siRNAs; B-cell lymphoma/leukemia-2 (Bcl-2) siRNA and vascular endothelial growth factor (VEGF) siRNA to glioblastoma cells effectively [165]. The findings of this study demonstrated the successful delivery of siRNAs to glioma cells while maintaining cytocompatibility. Moreover, the developed NPs enabled the effective transfection of both Bcl-2 and VEGF siRNAs leading to improved gene silencing via downregulating the expression of the respective proteins and hence increasing the antiproliferative effect [165].

Another study reported the fabrication of arginine-glycine-aspartic acid (RGD) peptide-surface-modified PAMAM dendrimers for the co-delivery of c-Myc-siRNAs and doxorubicin (DOX)-loaded selenium NPs aiming at suppressing glioblastoma progression. The RGD decorating the dendritic surface is a vital target moiety that could bind to the overexpressed αvβ3 integrin receptor on the surface of glioblastoma cells, thus maximizing cellular internalization while minimizing off-target effects. Besides, the designed nanosystem effectively passed the BBB to co-deliver both the siRNAs targeting the oncogenic mediators and the chemotherapeutic drug, leading to notable inhibitory effects on U251 cancer spheroids [166].

Due to its positive charge, chitosan (CS) NPs are another promising candidate for complexing the negatively charged siRNAs. In this context, CS-lipid NPs were designed to deliver anti-galectin-1 and anti-epidermal growth factor receptors-siRNAs in glioblastoma treatment because galectin-1 and EGFR are overexpressed in glioblastoma cells and are associated with the proliferation and invasion of cancer cells. The prepared NPs increased mice survival rates (due to the downregulation of EGFR and galectin-1) and improved chemosensitivity to the temozolomide anticancer drug [167]. Another study reported the rapid intra-nasal delivery of anti-galectin-1 siRNAs to the brain via complexing with CS NPs. Loading anti-galectin-1 siRNAs onto CS NPs increased its stability by preventing its hydrolysis by RNAase enzyme. The designed NPs could reduce more than 50% of galectin-1 in tumor-bearing mice. This promising study paves new avenues toward exploring the intranasal route in delivering RNAs to the brain [168].

Cationic lipid-based nanoemulsions (LPNs) are another excellent vehicle for the selective targeting of therapeutic RNAs due to their ability to form stable complexes with the anionic RNAs via ion-pair interactions [169].

Anotherstudy reported the efficient intranasal delivery of CD37-siRNAs via loading into cationic LNPs. The CD37 enzyme is overexpressed on the glioblastoma cell surface and increases adenosine synthesis, which is responsible for cancer cell proliferation and invasion. An in vitro cell viability study showed that the viability of cancer cells was reduced by more than 50%. In addition, the designed CD37-siRNAs LNPs were detected in the brain of the glioblastoma-bearing Wistar rats upon intranasal administration, demonstrating the prepared nanoemulsion’s capacity to pass the BBB. Moreover, it was shown that the treatment using the designed CD37-siRNAs LNPs had decreased tumor growth and adenosine levels by 60% and 95%, respectively. These findings highlighted the in vivo CD37 silencing ability of the prepared LNP when administered intranasally, which might represent a promising strategy in glioblastoma treatment [170]. In another study, angiopep-2-decorated cationic lipid-poly (lactic-co-glycolic acid) (PLGA) NPs were designed for the dual delivery of gefitinib (an EGFR tyrosine kinase inhibitor) and golgi phosphoprotein 3 (GOLPH3) siRNAs that would act as an effective combinatorial for anti-glioblastoma therapy. The decoration of the NPs with angiopep-2 was reported to enhance cancer cells’ targeting and promote the penetration of the BBB. It was revealed that the developed NPs successfully passed the BBB, while the delivered GOLPH3 siRNAs efficiently silenced the expression of GOLPH3 mRNA and increased the degradation of EGFR. In addition, it was shown that the targeted gefitinib significantly aborted the signaling through the EGFR. Thus, these combinatorial cationic lipid-PLGA NPs hold considerable promise as a potentially safe and potent anti-glioblastoma therapy [171].

Another recent study reported the formulation of siRNA-loaded lipopolyplexes formed of phospholipid 1,2-dipalmitoyl-sn-glycero-3-phosphocholine (DPPC) and polyethyleneimine (PEI) to the target signal transducer and activator of transcription 3 (STAT3), an essential signaling protein responsible for the development of glioblastoma. These study findings show that the prepared lipopolyplexes downregulate the expression of STAT3, reduce cancer cell proliferation, and remarkably increase survival rates in Tu2449 tumor-bearing mice [172].

Metallic and magnetic NPs are another platform for the targeted delivery of therapeutic RNAs to glioblastoma cells [173]. Metallic NPs offer high encapsulation efficiencies (EE) when they are used as carriers for RNAs. For instance, a recent study developed a combinatorial therapy for glioblastoma based on iron oxide NPs (IONPs) co-loaded with siRNAs-targeting glutathione peroxidase 4 (si-GPX4) and cisplatin (Cs, platinum-based anticancer drug). The designed NPs demonstrated high EEs of both siRNAs and Cs. In addition, the proposed NPs showed multimodal anticancer activities against U87MG, and P3#GBM brain cancer cells via (i) activating reduced nicotinamide adenine dinucleotide phosphate (NADPH) oxidase, leading to increasing hydrogen peroxide levels, (ii) increasing the intracellular iron levels which will eventually react with the hydrogen peroxide (Fenton reaction) forming reactive oxygen species (ROS) which induce ferroptosis, (iii) apoptosis caused by Cs, and (iv) si-GPX4 downregulating the expression of GPX4 and synergistically improving the anticancer effect. Moreover, the designed NPs showed minimal toxicity against normal human astrocytes [173,174]. Another recent study reported the enhancement of metallic NPs stability in siRNAs delivery and glioblastoma treatment via decoration with albumin [175]. In this regard, bovine serum albumin-stabilized manganese-based NPs were fabricated to efficiently deliver VEGF-siRNAs in brain tumor treatment. The prepared NPs were highly internalized and accumulated inside the cancer cells and demonstrated an improved anti-angiogenesis effect and a prolonged residence time. In addition, these novel NPs showed outstanding stability and exceptional biocompatibility, which warrant their huge potential for further clinical translation [175].

Silica NPs hold great potential in gene delivery because of their excellent biocompatibility, safety, and large surface areas, which lead to their ability to accommodate high RNA concentrations. A recent study reported the development of porous silica NPs coated with PEI for the targeted delivery of multidrug resistance-associated protein 1 (MRP-1)-siRNAs in the treatment of glioblastoma [176]. MRP-1 is overexpressed in glioblastoma and is responsible for the resistance to chemotherapy and the proliferation of tumor cells. The porous silica NPs coated with PEI demonstrated a high siRNA loading capacity (70%) and a sustained release behavior throughout a 48 h period. In addition, the prepared NPs showed a high cellular uptake and downregulated the expression of MRP-1 in glioblastoma cells (by about 30%), leading to cell cycle arrest and a remarkable reduction in proliferative activity [176]. All the presented studies are summarized in Table 6.

However, the nanodelivery of therapeutic RNAs still suffers some limitations which might hinder their clinical translation [177]. For instance, lipid-based NPs suffer from rapid release rates and short half-lives in the systemic circulation. Additionally, cationic polymers are toxic to non-cancerous cells, while mesoporous silica NPs exhibit liver toxicity. Moreover, gold NPs are expensive and showed rapid reticuloendothelial system clearance and fatal toxicities at certain nano-sizes [177]. Although the nanodelivery of therapeutic RNAs is very promising, still the major obstacle is finding specific agents to target and trim H_2_S production inside brain tumor cells.

On one hand, therapeutic RNAs targeting H_2_S synthesizing enzymes have surpassed all available pharmacological inhibitors available in the market in terms of efficiency and specificity; on the other hand, therapeutic RNAs have demonstrated a notable advancement in brain cancer therapy. Therefore, this highlights a real gap and hope for pediatric brain tumor patients. Further research should investigate the potential and effectiveness of nanocarriers carrying therapeutic RNAs specifically targeting H_2_S synthesizing enzymes in pediatric brain tumors validating its potential clinical translation. Especially, it is quite evident that loading therapeutic RNAs into various nanoplatforms has improved gene silencing efficiency, increased BBB penetration capacity, and overcome all the drawbacks of using individual therapeutic RNAs solely.

## 6. Conclusions and Future Recommendations

In conclusion, this review highlights the heterogeneous nature of brain tumors and the terrifying statistics of pediatric brain tumors. Several oncogenic signaling pathways have been characterized as powerful molecular engines underlying the aggressive nature of pediatric brain tumors. However, little is known about the involvement of H_2_S in the pathophysiology of pediatric brain tumors. In the current review, the authors have shed light onto most of the published literature focusing on the expression profile of H_2_S-synthesizing enzymes in different brain tumors. Studies focusing on the role of H_2_S in brain tumor cell lines, animal models, and human subjects have been extensively reviewed. It was generally clear that CBS is downregulated, whereas CSE and 3MST are upregulated and directly correlated with the brain tumor grade reaching the highest expression level at grade III; then, CSE and 3MST start to drop in their expression levels at later grades. This necessitates further investigations for analyzing the pattern of H_2_S levels in brain tissues during the malignant transformation process of pediatric brain tumors. It also highlights the importance of correlating the deregulation of H_2_S synthesizing enzymes with different molecular subtypes, age of children, metastasis, and aggressiveness of the disease. The authors also recommend the imperative need to give more attention to H_2_S-synthesizing enzymes as possible molecular targets for treating primary brain tumors, generally, and pediatric types, particularly. Last but not least, the authors in this review also provided an extensive review of different brain-delivery strategies and tactics that would help in the near clinical translation of the molecular targeting of H_2_S-synthesizing enzymes among pediatric brain tumor patients, creating a source of hope for pediatric brain tumor patients.

## Figures and Tables

**Figure 1 cancers-14-05244-f001:**
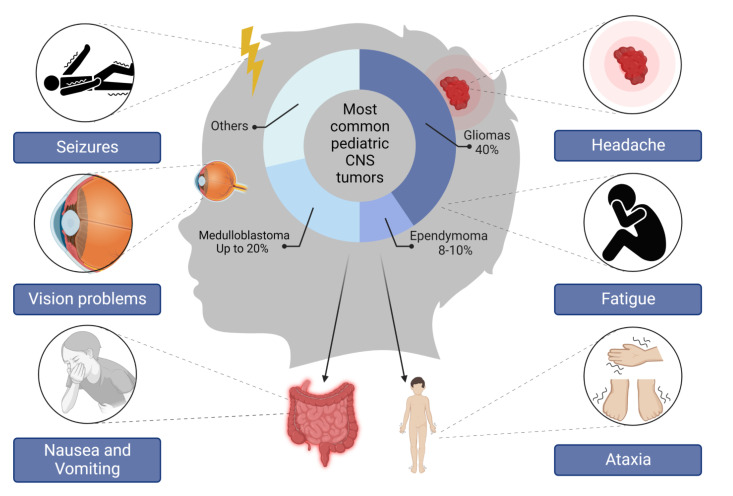
Brain tumor-associated clinical symptoms and the most common types in the pediatric population.

**Figure 2 cancers-14-05244-f002:**
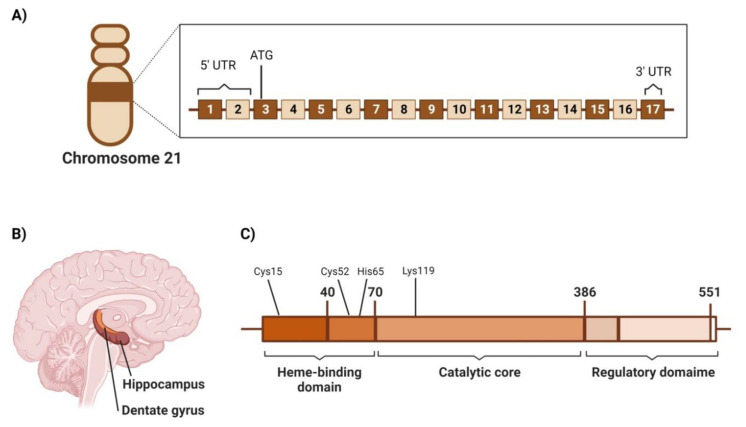
CBS chromosomal location, localization in brain, and molecular domain structure. (**A**) Human CBS gene structure including exons and untranslated regions. Numberings are relative to the translation start site ATG. (**B**) Spatial distribution of CBS enzyme in the human brain (**C**) The modular domain structure of human CBS enzyme.

**Figure 3 cancers-14-05244-f003:**
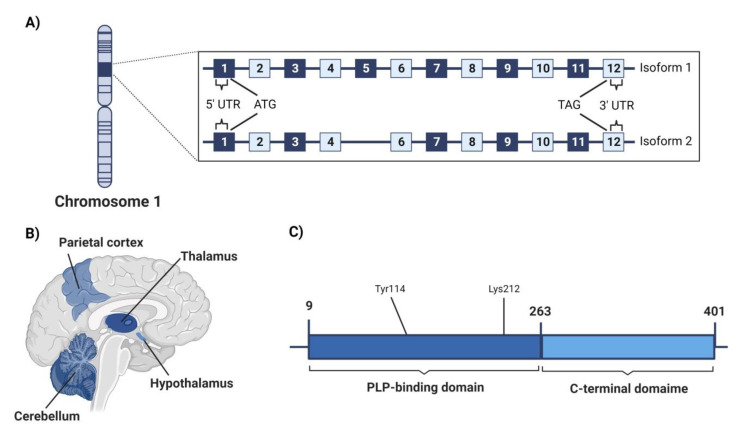
CSE chromosomal location, localization in brain, and molecular domain structure. (**A**) Structure of human CTH gene which encodes for CSE enzymes including exons and untranslated regions. Numberings are relative to the translation start and end sites, ATG and TAG, respectively. (**B**) Spatial distribution of CSE enzyme in the human brain. (**C**) The modular domain structure of one monomer of human CSE enzyme.

**Figure 4 cancers-14-05244-f004:**
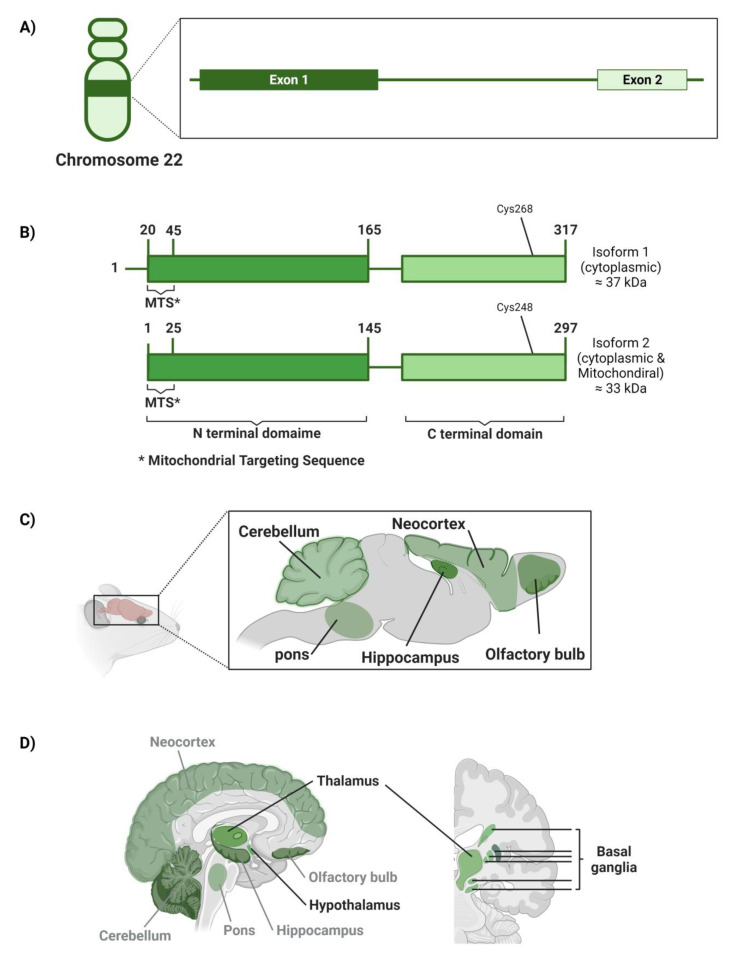
The 3MST chromosomal location, localization in the brain, and molecular domain structure. (**A**) The structure of the human *MPST* gene encodes for the 3MST enzyme, including exons and introns. Numberings are relative to the translation start sites. (**B**) The modular domain structure of one monomer of human 3MST enzyme. (**C**) The spatial distribution of 3MST enzyme in mouse brain. (**D**) Spatial distribution of 3MST enzyme in the human brain; sagittal cut (on the **left**) shows thalamus and hypothalamus location and other corresponding areas for mouse brain (gray); coronal cut (on the **right**) shows basal ganglia location.

**Table 1 cancers-14-05244-t001:** Most prevalent pediatric CNS tumors with their prevalence, average age, prognosis, genetic alterations, standard treatment strategies, and suspected outcomes.

Tumor	% Prevalence in Children	Age (Years)	WHO Grade	5 Years Overall Survival	Prognosis	Predisposed Genetic Mutation	Standard Treatment	Suspected Outcome	Ref.
(I) Medulloblastoma									
1 WNT	2%	>3 years old	-	95%	Very good	- CTNNB1 mutations - Monosomy 6	Surgery, Radiation, chemotherapy	- Improved outcomes - Posterior fossa syndrome - Neurocognitive deficits following RT - Secondary malignancies	[7,9,74,75,76]
2 SHH	6%	All over Age spectrum	-	50–75% with metastasis or <50% with TP53 mutation	Poor	- TP53 mutation - MYNC amplification - PTCH1/SMO/SUFU mutation	- Surgery, Radiation, chemotherapy - Small molecule inhibitors: erismodegib/sonidegib/vismodegib	Small molecule inhibitors harbor better results for adults harbor than children	[7,9,77,78,79]
3 Group 3	5%	Mostly infants and toddlers	-	50%	Extremely poor	- MYC amplification - PVT1-MYC fusion	Surgery, Radiation, chemotherapy	Carries worst prognosis with current therapies	[7,9]
4 Group 4	7%	Mostly children and teenagers	-	90% or 50–75% according to mutation	Intermediate	- CDK6 amplification - Isochromosome 17q - SNCAIP duplication	Surgery, Radiation, chemotherapy	No breakthrough in treatment options	[7,9,77]
(II) Gliomas									
1 High-Grade Glioma									
a Anaplastic astrocytoma (AA)	1.5%	0–19	III	32%	Poor prognosis	Thalamic tumors have FGFR1 mutations + H3.3 point mutations	- Radiation is the mainstay of therapy as well as surgical resection. - Adjuvant therapy with CCNU, vincristine, and prednisone. - Erlotinib, Imatinib, and bevacizumab were used as targeted therapies	- High-dose chemotherapy shows small survival benefits and higher toxicity levels - Chemotherapy benefits are controversial - Minimal benefits of targeted therapies against pediatric AA and GBM	[7,9,80]
b Glioblastoma (GBM)	2.9%	0–19	IV	18%	Poor prognosis	DNA methylations			
2 Low-Grade Glioma									
a Pilocytic astrocytoma	15.6%	0–14	I	97%	Excellent	BRAF V600E	- RT is the first line of treatment - Surgical resection - carboplatin and vincristine	- Induced brain injuries of RT at young age eliminate it. - High morbidity and mortality rates, functional compromise when the tumor is in close proximity to vital structures (total resection not possible) - Tumor growth within 3 years of chemotherapy initiation	[7,9,81,82]
(III) Ependymoma									
1 PF-EPN-A	N/A	Infants and young children	II	N/A	Worst prognosis	Balanced Genome	- Surgical resection followed by radiation therapy - Chemotherapy in an aim to delay the use of radiation	- 50% mortality from the disease - Difficulty to resect - Survival rate 0–11% after subtotal resection - High recurrence rate - Radiation in children <3 years is controversial - Treatment-induced diseases ex: stroke, secondary malignancies, and neurocognitive deficits.	[7,9,83]
2 ST-EPN-RELA	N/A	All over age spectrum	III	N/A	Worst prognosis	Chromotripsis, RELA fusion			

**Table 2 cancers-14-05244-t002:** Conventional therapeutic strategies for CNS tumors before and after 2016.

Tumor Type	Before 2016		After 2016	
	Standard Treatment	Outcomes	New Therapeutic Approaches	Outcomes
Medulloblastoma	Surgery, radiation, and chemotherapy	- Posterior fossa syndrome. - Neurocognitive impairment following RT. - Secondary malignancies.	- RT and chemotherapy dose reduction. - SMO inhibition for SHH. - Small molecule inhibitors. - HDAC and bromodomain inhibitors.	Better therapeutic effects in adults versus children.
Glioma	- RT and surgical resection. - Chemotherapy.	- RT-induced injuries. - High toxicity. - High morbidity risk.	Targeted therapies such as erlotinib and ematinib.	Minimal benefits for pediatric AA and GBM.
Ependymoma	- Surgical resection followed by radiation. - Chemotherapy.	- High recurrence rate. - Radiation in children <3 years is controversial. - Treatment-induced dysfunctions.	Epigenetic demethylation drugs.	Under clinical research

**Table 3 cancers-14-05244-t003:** Mechanisms of halting CBS activity.

Class	Inhibitor	Reference
Pharmacological inhibitors	Aminooxyacetic acid (AOAA)	[105]
	Copper diethyldithiocarbamate [Cu(DDC)_2_]—disulfiram metabolite	[106]
	Benserazide	[107]
miRNAs	miR-559	[108]
	miR-125b-5p	[109]
	miR-24-3p	[110]
	miR-4317	[111]
shRNAs	shCBS#07	[104]

**Table 4 cancers-14-05244-t004:** Mechanisms of halting CSE activity.

Class	Inhibitor	Reference
Pharmacological inhibitors	propargylglycine (PAG)	[119]
	Aminooxyacetic acid (AOAA)	[105]
	aurintricarboxylic acid (NSC4056)	[120]
	β-cyanoalanine (BCA)	[105]
	L-aminoethoxyvinylglycine (AVG)	[105]
	Hydroxylamine	[105]
	I157172	[121]
miRNAs	miR-21	[122]
	miR-186	[123]
	miR-30	[124]
	miR-4317	[111]

**Table 5 cancers-14-05244-t005:** Mechanisms of halting 3MST activity.

Class	Inhibitor	Reference
Pharmacological inhibitors	HMPSNE: 2-[(4-hydroxy-6-methylpyrimidin-2-yl) sulfanyl]-1-(naphthalen-1-yl)ethan-1-one	[135]
	2-(2-Naphthalen-1-yl-2-oxo-ethylsulfanyl)-3H-pyrimidin-4-one	[136]

**Table 6 cancers-14-05244-t006:** Nanocarriers for therapeutic RNAs target delivery in brain tumors.

Nanoplatform	Therapeutic RNA Target	Surface Decoration	Chemotherapeutic Agent	Advantages	Ref
Cationic liposomes	VEGF	Angiopep-2 and neuro pilin-1 receptor	docetaxel	- Suppressed viability and angiogenesis of brain glioblastoma cells - Downregulated HIF-1α expression and CD31-positive tumor vessels - Modulated tumor microenvironment	[161]
Liposomes	FTH1	-	-	- Decreased cell viability of cancer cells - Increased mitochondrial damage. - Improved radiosensitivity.	[162]
Liposomes	CD13 receptors	Asn-Gly-Arg (NGR) peptide-targeting ligand	-	- Prolonged rat survival rate - 9-fold increase in shRNA delivery to rat gliomas	[163]
β-cyclodextrin (β-CD)-coated PAMAM	Bcl-2 VEGF	-	-	- Improved gene silencing. - Enhanced antiproliferative effect.	[165]
PAMAM dendrimers	c-Myc	arginine-glycine-aspartic acid (RGD) peptide	doxorubicin (DOX) loaded selenium NPs	- Maximizing cancer cells’ internalization. - Synergistic anticancer effects. - Enhanced permeability to BBB.	[166]
Chitosan-Lipids NPs	Galectin-1 EGFR	-	-	- Increased mice survival rates. -Improved chemosensitivity to temozolomide anticancer drug.	[167]
Chitosan NPs	Galectin-1	-	-	- Innovative intra-nasal delivery to brain tumors. - Improved stability. - Reduced > 50% of galectin-1 in tumor-bearing mice	[168]
Lipid-based nanoemulsions	CD37	-	-	- Reduced the viability of cancer cells by >50%. - Decreased tumor growth and adenosine levels by 60% and 95%, respectively. - Improved capability to pass BBB via the intra-nasal route.	[170]
Cationic lipid-poly (lactic-co-glycolic acid) (PLGA) NPs	Golgi phosphoprotein 3 (GOLPH3)	Angiopep-2	Gefitinib	- Improved brain cancer cells’ targeting. - Promoted the penetration of the BBB. - Silenced the expression of GOLPH3 mRNA. - Increased the degradation of EGFR. - Biocompatible.	[171]
Lipopolyplexes	STAT3	-		- Downregulated the expression of Stat3. - Reduced cancer cell proliferation. - Prolonged survival rates in Tu2449 tumor-bearing mice.	[172]
Iron oxide NPs (IONPs)	GPX4	-	Cisplatin	- Multimodal anticancer activities against U87MG and P3#GBM cells via ferroptosis and apoptosis. - Minimal toxicity against normal human astrocytes	[173]
Manganese-based NPs	VEGF	Bovine serum albumin	-	- Improved stability. - Biocompatibility. - High internalization and accumulation inside the cancer cells. - Improved anti-angiogenesis effect. - Prolonged blood circulation duration.	[175]
Porous silica NPs	MRP-1		PEI	- High siRNA loading capacity of 70%. - Sustained release behavior. - High cellular uptake. - Downregulated the expression of MRP-1 in glioblastoma cells (by about 30. - Notable reduction in proliferative activity	[176]

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
