# Peer review of "Molecular Engines, Therapeutic Targets, and Challenges in Pediatric Brain Tumors: A Special Emphasis on Hydrogen Sulfide and RNA-Based Nano-Delivery"

_cancers, 2022, doi:10.3390/cancers14215244_

Round 1
Reviewer 1 Report
This is a very well-written and comprehensive review article. It discussed the different types of pediatric brain tumors and the established therapy for CNS tumors. More importantly, focusing on Hydrogen Sulfide (H2S), the manuscript discussed features of enzymes which synthesize H2S from the aspects of the genetic location, structure, expression patterns, and function in brain tumors. Last but not least, it discussed the therapeutic approaches which can be used and delivered to brain tumors.
The manuscript is very well-written and structured. I therefore only have two minor points to point out:
1. Writing of H2S needs to be consistent through out the manuscript.
2. Lines 624-626 have grammar issue.
It can be accepted once the minor issues are fixed.
Author Response
Dear Reviewers,
We are thankful for your thorough review of our manuscript which aims for the improvement and scientific enrichment of our paper. All the issues noted by reviewers have been taken into consideration and extensively revised. The comments and inquiries are shown in Bold and the replies are shown in Italic.
Reviewer 1:
This is a very well-written and comprehensive review article. It discussed the different types of pediatric brain tumors and the established therapy for CNS tumors. More importantly, focusing on Hydrogen Sulfide (H2S), the manuscript discussed features of enzymes that synthesize H2S from the aspects of the genetic location, structure, expression patterns, and function in brain tumors. Last but not least, it discussed the therapeutic approaches which can be used and delivered to brain tumors.
The manuscript is very well-written and structured. I therefore only have two minor points to point out:
We are extremely thankful for the reviewer’s feedback and comments. We are very glad that we met your expectations concerning the quality of the review.
- Writing of H2S needs to be consistent throughout the manuscript.
We are thankful for the reviewer’s comment regarding this point. This has been taken into consideration while meticulously revising the manuscript and preparing the revised version. All the required changes have been highlighted in the revised manuscript.
- Lines 624-626 have grammar issue.
According to the reviewer’s comment, this section has been revised, re-structured and all typos and grammatical mistakes were revised throughout the manuscript.
Reviewer 2 Report
This is an excellent review article focusing on hydrogen sulfide and its role in brain tumor development and suppression, available technologies, and future directions. The authors start with the significance of hydrogen sulfide in brain tumor tissues and the limitations of currently available pharmacological inhibitors. The authors make it interesting by exploring the potential route of overcoming these challenges through the utilization of different nano-delivery systems currently in research. This manuscript is an interesting summary worth reading.
Minor comments:
- In section 5.2 the authors list currently available research on nano-delivery therapeutics of RNA. In the excitement of explaining these technologies, they miss out on explaining their limitations too and, in that way, please highlight the need for exploration of different targets like hydrogen sulfide because if targeting various molecules with the technology worked then is there a need to specifically focus on hydrogen sulfide?
-If possible, include DIPG since it is prevalent among children with an extremely low survival rate. I believe it will increase the impact of the article.
-Section starting from line 209 together with Figure 1 fits better at the end of the Introduction section, just before covering different types of brain tumors.
Author Response
Dear Reviewers,
We are thankful for your thorough review of our manuscript which aims for the improvement and scientific enrichment of our paper. All the issues noted by reviewers have been taken into consideration and extensively revised. The comments and inquiries are shown in Bold and the replies are shown in Italic.
This is an excellent review article focusing on hydrogen sulfide and its role in brain tumor development and suppression, available technologies, and future directions. The authors start with the significance of hydrogen sulfide in brain tumor tissues and the limitations of currently available pharmacological inhibitors. The authors make it interesting by exploring the potential route of overcoming these challenges through the utilization of different nano-delivery systems currently in research. This manuscript is an interesting summary worth reading.
Minor comments:
- In section 5.2 the authors list currently available research on nano-delivery therapeutics of RNA. In the excitement of explaining these technologies, they miss out on explaining their limitations too and, in that way, please highlight the need for exploration of different targets like hydrogen sulfide because if targeting various molecules with the technology worked then is there a need to specifically focus on hydrogen sulfide?
We are thankful for the reviewer’s comment for raising this important point. A separate paragraph has been added to the revised manuscript, lines (634-641) highlighting the limitations and the urgent need to explore different agents to potentially target and trim hydrogen sulfide production inside brain tumor cells.
-If possible, include DIPG since it is prevalent among children with an extremely low survival rate. I believe it will increase the impact of the article.
We are thankful for the reviewer’s comment regarding this point. A separate paragraph has been added to the revised manuscript, lines (190-196) highlighting the characteristics of DIPG subtype among children and its molecular pathogenesis.
-Section starting from line 209 together with Figure 1 fits better at the end of the Introduction section, just before covering different types of brain tumors.
According to the reviewer's comment, this section has been added to the introduction section lines 82-100 in the revised manuscript.
Reviewer 3 Report
I commend the authors for their extensive compilation of literature on molecular alterations in pediatric brain tumors.
The introduction is rather extensive and generic – most of it could be part of any essay on (pediatric) brain tumors. Maybe consider a shorter, but more specific introduction explaining what is known about hydrogen sulfide, its role in cancer and why it is interesting to consider its role in brain tumors (and why specifically in pediatric brain tumors). In fact, the title is somewhat misleading, as the manuscript is not primarily on the role of hydrogen sulfide in pediatric brain tumors, but rather on molecular signalling pathway alterations in pediatric brain tumors in general.
Line 160: “the discovery of the reason behind low-grade glioma”: That is misleading. It does not appear likely that there is THE one “reason behind low grade glioma” in the first place.
The prevalences in Table 1 are confusing. I presume that for medulloblastoma types, the ratio to all medulloblastomas is given, but this is inconsistent with the prevalences given for gliomas
There are also many wording / language issues, e.g.:
Line 36: “brain delivery strategies for the sake of achieving higher therapeutic privilege”: therapeutic privilege usually refers to the withholding of information by the clinician from the patient
Line 48 (“15-20% of all other…malignancies”): what “other” malignancies?
Line 69: (“advances in the molecular pathogenesis”): advances in the understanding of..
Line 435-437: “Given the promising involvement of H2S in brain tumor pathophysiology and the importance of investigating as a therapeutic target, H2S has shown a bell-shaped pharmacological character in different cancer types.”
Line 614/615: “..signaling pathways have been characterized underneath the aggressive nature of pediatric brain tumors.”
Line 621/622: “..both starts to drop in their expression levels at later grades.”
These are just a few of quite a lot of examples. Many sentences are hard to understand, if at all. I would suggest to do a thorough language check.
Author Response
Dear Reviewers,
We are thankful for your thorough review of our manuscript which aims for the improvement and scientific enrichment of our paper. All the issues noted by reviewers have been taken into consideration and extensively revised. The comments and inquiries are shown in Bold and the replies are shown in Italic.
I commend the authors for their extensive compilation of literature on molecular alterations in pediatric brain tumors.
We are extremely thankful for the reviewer’s feedback and comments. All mentioned comments have been taken into consideration while meticulously revising the manuscript and preparing the revised version. The required changes have been highlighted in the revised manuscript.
The introduction is rather extensive and generic – most of it could be part of any essay on (pediatric) brain tumors. Maybe consider a shorter, but more specific introduction explaining what is known about hydrogen sulfide, its role in cancer and why it is interesting to consider its role in brain tumors (and why specifically in pediatric brain tumors).
According to the reviewer's comment, the introduction section in the revised manuscript has been restructured where a special emphasis of several subtypes/subclasses of pediatric brain tumors were highlighted together with their associated symptoms. Also a more detailed emphasis on the hydrogen sulfide as a novel player in the oncological context has been highlighted.
In fact, the title is somewhat misleading, as the manuscript is not primarily on the role of hydrogen sulfide in pediatric brain tumors, but rather on molecular signalling pathway alterations in pediatric brain tumors in general.
According to the reviewer's comment, the title of the manuscript has been amended in the revised manuscript into " Molecular Engines, Therapeutic Targets and Challenges in Pediatric Brain Tumors: A Special Emphasis on Hydrogen Sulphide"
Line 160: “the discovery of the reason behind low-grade glioma”: That is misleading. It does not appear likely that there is THE one “reason behind low grade glioma” in the first place.
According to the reviewer's comment, this sentence has been revised and corrected in the revised manuscript
The prevalence in Table 1 are confusing. I presume that for medulloblastoma types, the ratio to all medulloblastomas is given, but this is inconsistent with the prevalences given for gliomas
According to the reviewer's comment, this section has been revised and written in a more reader-friendly format in the revised manuscript. All reported prevalence percentages mentioned in the table were percentages among total pediatric tumors.
There are also many wording / language issues, e.g.:
Line 36: “brain delivery strategies for the sake of achieving higher therapeutic privilege”: therapeutic privilege usually refers to the withholding of information by the clinician from the patient
We are thankful for the reviewer's comment raising this point; this phrase has been re-structured in the revised manuscript to be clearer.
Line 48 (“15-20% of all other…malignancies”): what “other” malignancies?
According to the reviewer's comment, this phrase has been edited in the revised manuscript to be clearer.
Line 69: (“advances in the molecular pathogenesis”): advances in the understanding of.
According to the reviewer's comment, this phrase has been corrected in the revised manuscript to be clearer.
Line 435-437: “Given the promising involvement of H2S in brain tumor pathophysiology and the importance of investigating as a therapeutic target, H2S has shown a bell-shaped pharmacological character in different cancer types.”
According to the reviewer's comment, this phrase has been re-structured and edited in the revised manuscript to be clearer.
Line 614/615: “..signaling pathways have been characterized underneath the aggressive nature of pediatric brain tumors.”
According to the reviewer's comment, this phrase has been edited and corrected in the revised manuscript to be clearer.
Line 621/622: “..both starts to drop in their expression levels at later grades.”
According to the reviewer's comment, this phrase has been edited and corrected in the revised manuscript to be clearer.
These are just a few of quite a lot of examples. Many sentences are hard to understand, if at all. I would suggest to do a thorough language check.
According to the reviewer's comment, this has been taken into consideration while meticulously revising the manuscript and preparing the revised version of our manuscript. All typos and grammatical mistakes have been revised thoroughly and edited in the revised manuscript.
Round 2
Reviewer 3 Report
I thank the authors for taking my suggestions into account. However, I have to point out that the sentences I quoted in my review report were just examples, and numerous linguistic issues have remained in the current version. It would be too much to list them all. These are just a few examples from the first sentences of the "simple summary" which is quite difficult to read:
"Pediatric brain tumors represents a formidable challenge in the field of oncology"
"eachone"
"Although, the great efforts dedicated to portray the involved signaling pathways driving the pediatric brain tumors." => this is not a complete sentence
"In this review, the authors shed the light onto novel therpauetic targets tailored for several sub-types of pediatric brain tumors and spots out the limitations for such therapeutic approaches."